# *Preference Leakage*: A Contamination Problem in LLM-as-a-judge

**Dawei Li**[*1]**, Renliang Sun**[*2]**, Yue Huang**[3]**, Ming Zhong**[4]**, Bohan Jiang**[1]**,
**Jiawei Han**[4]**, Xiangliang Zhang**[3]**, Wei Wang**[2]**, Huan Liu**[1]
[1]Arizona State University, [2]University of California, Los Angeles,
[3]University of Notre Dame, [4]University of Illinois Urbana-Champaign
`daweili5@asu.edu`

## ABSTRACT

Large Language Models (LLMs) as judges and LLM-based data synthesis have emerged as two fundamental LLM-driven data annotation methods in model development. While their combination significantly enhances the efficiency of model training and evaluation, little attention has been given to the potential contamination brought by this new model development paradigm. In this work, we expose preference leakage, a contamination problem in LLM-as-a-judge caused by the relatedness between the synthetic data generators and LLM-based evaluators. To study this issue, we first define three common relatednesses between the data generator LLM and the judge LLM: being the same model, having an inheritance relationship, and belonging to the same model family. Through extensive experiments, we empirically confirm the bias of judges towards their related student models caused by preference leakage across multiple LLM baselines and benchmarks. Further analysis suggests that preference leakage is a pervasive and real-world problem that is harder to detect compared to previously identified biases in LLM-as-a-judge scenarios. All of these findings imply that preference leakage is a widespread and challenging problem in the area of LLM-as-a-judge. We release all codes and data at: `https://github.com/David-Li0406/Preference-Leakage`[1].

## 1 INTRODUCTION

Recent advancements in Large Language Models (LLMs) Achiam et al. (2023); Jaech et al. (2024); Tong et al. (2024); Zhang et al. (2024a); Tan et al. (2024a; 2025); Huang et al. (2026) have empowered various downstream tasks and applications. However, this also poses substantial challenges to the automatic evaluation of these models. Representatively, LLM-based AI agents' focus transfer from traditional natural language processing tasks Yang et al. (2023); Zhang et al. (2023) to real-world Liu et al. (2023b); Huang et al. (2023), open-ended response generation Wu et al. (2024), which greatly limits the applicability of traditional n-gram matching methods (e.g., BLEU Papineni et al. (2002) and ROUGE Lin (2004)) Liu et al. (2016); Reiter (2018) or model-based evaluators Zhang et al. (2020); Zhong et al. (2022) for evaluation.

To address these challenges, the paradigm of LLM-as-a-judge Zheng et al. (2023a); Li et al. (2024a); Jiang et al. (2024a); Zhong et al. (2024); Li et al. (2025a); Qu et al. (2025) has been proposed, designed to leverage LLM as evaluators to assess response quality. By combining powerful LLMs with well-designed prompting strategies, LLM-as-a-judge enables human-like evaluation of long-form and open-ended generation in a more cost-efficient and scalable manner. However, recent studies point out some weaknesses of such an assessment. For instance, Ye et al. (2024) explores various biases and vulnerabilities of LLM-as-a-judge, highlighting the importance of developing a reliable and fair LLM-based evaluation system.

In this work, we aim to highlight a subtle yet critical bias in LLM-as-a-Judge: *Preference Leakage*. This issue arises when *the LLMs used for data generation and evaluation are closely related, causing*

---

[*] Equal contribution.

[1]More resources on LLM-as-a-judge are on the website: `https://llm-as-a-judge.github.io/`

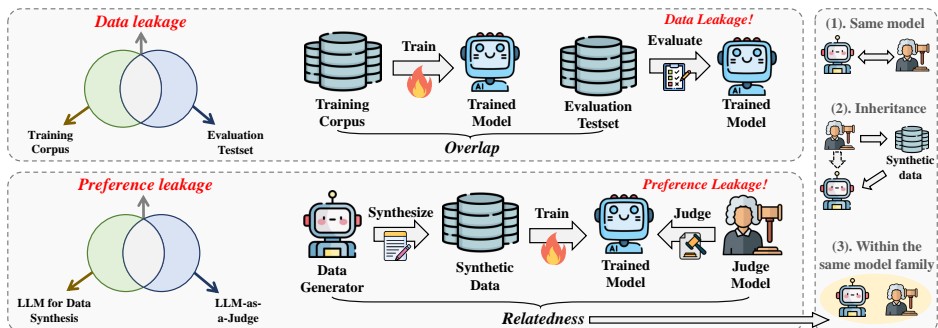

Figure 1: Overview of preference leakage. We make a comparison between data leakage and preference leakage and present three types of relatedness: being the same model, having an inheritance relationship and belonging to the same model family.

*the preference of the LLM evaluators to leak to the student models through synthetic data and thus inflating the evaluation score* (as illustrated in Figure 1). Synthetic data generated by LLMs Gan et al. (2023); Tan et al. (2024b); Li et al. (2024b;c) has become a cornerstone of model training Lee et al. (2025). When combined with LLM-as-a-Judge, they offer significant efficiency gains in model development. However, limited attention has been given to the potential contamination that occurs when the generator and evaluator LLMs share a close relationship. During our preliminary study, we find this issue is particularly pervasive in popular LLM-as-a-judge benchmarks (e.g., AlpacaEval 2.0 Dubois et al. (2024) and Arena-Hard Li et al. (2024e)) and LLM-relevant studies (more details can be found in Appendix B), due to the common reliance on the most advanced LLMs, such as GPT-4 Achiam et al. (2023), for both data synthesis and evaluation to ensure the highest quality outputs. In our work, we reveal this relatedness—akin to the overlap between training data and evaluation sets in traditional data contamination—would introduce a systematic bias of judge LLMs towards their related student models (i.e., the model distilled by the data generator which is related to the judge). Compared to other biases in LLM-as-a-Judge, such as length bias or egocentric bias Ye et al. (2024); Panickssery et al. (2024), preference leakage is subtler and more challenging to detect, especially given that most LLMs do not disclose their training data.

To investigate and reveal the preference leakage problem, we first define three relatednesses between data generator LLM and judge LLM: being the same model, having an inheritance relationship, and belonging to the same model family. Each of these scenarios is commonly encountered in real-world applications. Then, we pose and answer three core research questions about preference leakage:

- **RQ1: Does preference leakage introduce systematic biases in LLM-based evaluation?** To answer it, we conduct experiments with various LLM baselines in two widely recognized LLM-as-a-judge benchmarks, also introduce the preference leakage score to quantify the bias caused by preference leakage. The analysis results suggest an obvious bias of judging LLMs toward their related student models due to preference leakage.
- **RQ2: What is the severity of preference leakage under various scenarios?** We conduct experiments under various data mixing strategies, relatedness settings, tuning techniques and real-world applications to address it, finding that preference leakage consistently affects judge LLMs. Moreover, the severity of preference leakage correlates with the degree of relatedness between the data generator and LLM judges, as well as the proportion of synthetic data.
- **RQ3: What are the underlying mechanisms causing preference leakage?** For this question, we analyze LLMs' recognition capabilities on their related student models' generation as well as the distribution of bias across different question types and judgment dimensions. The analysis reveals that preference leakage is a subtle, hard-to-detect issue for the LLM evaluators, particularly affecting subjective questions and judgment dimensions.

To summarize, our contributions in this work are as follows:

- For the first time, we introduce preference leakage, a contamination issue arising from the relatedness between the data generator and judge LLMs.
- We conduct extensive experiments across various LLMs and benchmarks to study how and to what extent the potential bias brought by preference leakage influences judgment.

- Our further analysis reveals that preference leakage is prevalent in diverse scenarios and difficult for judge LLMs to detect, providing valuable insights for future research on this challenging issue.

## 2 RELATED WORK

**LLM-as-a-Judge.** LLM-as-a-Judge, introduced by Zheng et al. (2023a), leverages LLMs to automatically evaluate responses and assign rewards. This approach has gained widespread adoption in areas such as model alignment Zhang et al. (2024d) and benchmarking Liu et al. (2023a); Zhang et al. (2024b); Gao et al. (2023); Zhong et al. (2024); Beigi et al. (2024); Li et al. (2026), driving significant progress in the field. Building on this concept, Zhuge et al. (2024) proposed Agent-as-a-Judge, where agentic systems are employed to evaluate other agentic systems. Additionally, Prometheus, a series of open-source LLMs tailored for LLM-as-a-Judge Kim et al. (2023; 2024), addresses the prohibitive costs associated with proprietary models, further democratizing the technology.

Despite its promising potential, recent studies have highlighted the vulnerabilities and biases of LLM-as-a-Judge Zheng et al. (2023a); Ye et al. (2024); Koo et al. (2023); Chen et al. (2024); Zheng et al. (2023a); Huang et al. (2024); Thakur et al. (2024); Shi et al. (2024). Among these, egocentric bias, where LLM evaluators tend to favor their generations Koo et al. (2024); Liu et al. (2024b); Wataoka et al. (2024); Xu et al. (2024b); Rando et al. (2025); Panickssery et al. (2024); Chen et al. (2025), is most closely related to the preference leakage proposed in this work.

However, in contrast to the relatively straightforward setting of egocentric bias, preference leakage presents a more complex and dynamic challenge. It can arise from various types of relatedness between data-generating and evaluating LLMs, as well as the intricate flow of synthetic data among modern LLMs Tan et al. (2024b). Moreover, detecting preference leakage is also more challenging, given LLMs often do not disclose their training data and the difficulty in distillation quantification Wadhwa et al. (2025); Lee et al. (2025).

**Data Leakage.** The possible overlap between training data and evaluation benchmarks has become a central issue, since LLMs are usually trained on extensive web corpora Dodge et al. (2021). This phenomenon, known as data leakage, can artificially improve the performance of LLMs and undermine the reliability of the assessment Deng et al. (2024a); Jiang et al. (2024b); Li et al. (2025b). Several researchers have proposed methods to detect and mitigate data contamination. Deng et al. (2024b) proposed a retrieval-based approach to assess the degree of overlap between pre-training text and benchmark data. Golchin & Surdeanu (2023) have developed "guided instruction" to flag contaminated instances. Dong et al. (2024b) proposed the CDD method to identify peaks in the output distribution to detect data contamination. Several studies analyze data leakage for specific LLMs Balloccu et al. (2024); Xu et al. (2024a) and report contamination such as cross-language contamination Yao et al. (2024) and task contamination Li & Flanigan (2024) that can evade traditional detection methods. To address data contamination issues, Ni et al. (2024) have used web user query detection and benchmark mixture. White et al. (2024) use the most recent information to update the problem.

## 3 PREFERENCE LEAKAGE

### 3.1 LLMS AS ORACLES: A NEW AVENUE FOR CONTAMINATION

With the advent of LLMs, these models are increasingly employed as "oracles" in various scenarios: for both synthetic data generation ($M_G$) and employed as evaluators ($M_J$) to automate the assessment. While these approaches enhance scalability and efficiency, they also introduce potential risks. Specifically, if the LLM used for data generation ($M_G$) and the LLM used for evaluation ($M_J$) are not independent, a new contamination—preference leakage—can emerge, biasing evaluation outcomes.

### 3.2 DEFINING PREFERENCE LEAKAGE IN LLM-BASED EVALUATION

Formally, to define preference leakage, we consider the following entities in models development:

- **Data Generator LLM,** $M_G$, defining a conditional distribution $P_{M_G}(y|x)$ for generating an output $y$ given a prompt $x$, forming the synthetic dataset $D_{syn}$ for student LLMs training.

- **Student LLM,** $M_S$, trained on data generated by $M_G$, producing an output distribution $P_{M_S}(y|x)$.
- **Judge LLM,** $M_J$, providing a scoring function $S_{M_J}(y|x)$ that assesses output $y$ for prompt $x$.

Preference leakage occurs when the evaluation score assigned by $M_J$ to $M_S$'s outputs is inflated due to an underlying relatedness between $M_G$ and $M_J$. This implies that $M_J$ may favor outputs from $M_S$ not solely based on their intrinsic quality, but because they exhibit spurious features (e.g., style, format, wording) inherited from $M_G$, to which $M_J$ is predisposed due to this relatedness:

$$E_{x,y_S \sim P_{M_S}}[S_{M_J}(y_S|x)|M_G \sim_{rel} M_J] > E_{x,y_S \sim P_{M_S}}[S_{M_{J'}}(y_S|x)|M_G \not\sim_{rel} M_{J'}], \qquad (1)$$

where $y_S$ are outputs from $M_S$. The relation $M_G \sim_{rel} M_J$ denotes that judge $M_J$ is related to $M_G$, while $M_G \not\sim_{rel} M_{J'}$ denotes that an alternative judge $M_{J'}$ is not related to $M_G$ and possess comparable intrinsic quality assessment capabilities to $M_J$. The expectation is taken over the input distribution $\mathcal{X}$ and the trained Student LLM's output distribution $P_{M_S}$.

### 3.3 TYPE OF LLM "RELATEDNESS"

The condition $M_G \sim_{rel} M_J$ in Equation 1 encapsulates several ways the Data Generator LLM and Judge LLM can be interconnected. We identify three common types in the real world:

- **Being the Same Model:** The most direct form of relatedness occurs when the Data Generator LLM and the Judge LLM are the exact same model instance:

$$M_G \equiv M_J. \qquad (2)$$

  In this scenario, the inherent preferences in the model that shape its generative distribution $P_{M_G}(y|x)$ are precisely the same as those guiding its evaluation via the scoring function $S_{M_G}(y|x)$.
- **Inheritance Relationship:** One model's development is directly based on another, either by fine-tuning the existing model or by training a new model on the other's outputs, for instance:

$$M_J \leftarrow \text{FineTune}(M_G, D_{train}) \quad \text{or} \quad M_J \leftarrow \text{FineTune}(M_{base}, D_{syn_G}), \qquad (3)$$

  where $D_{train}$ represents general training data used to adapt $M_G$ into $M_J$, $M_{base}$ is a base model, and $D_{syn_G}$ denotes synthetic data generated by $M_G$. This type of relationship is bidirectional; $M_G$ can similarly inherit from $M_J$ through analogous processes. In such cases, the descendant model is likely to internalize and thus favor the preferences, styles, or biases of its progenitor.
- **Within the Same Model Family:** The Data Generator LLM $M_G$ and Judge LLM $M_J$ belong to the same model family (e.g., different versions or sizes of GPT). Models within such a family typically share a common architectural blueprint ($A_X$) and are often developed from foundational models pre-trained on substantially overlapping datasets ($D_X$). This shared foundation ($A_X, D_X$) would lead to correlated preferences and systemic biases characteristic of the common origin:

$$M_k \in \text{Family}(A_X, D_X) \quad \text{for } k \in \{G, J\}. \qquad (4)$$

## 4 MAIN EXPERIMENT

### 4.1 EXPERIMENT SETUP

**Models.** We choose three powerful LLMs as data generator/ judge models. They are GPT-4o-2024-11-20 Achiam et al. (2023), Gemini-1.5-flash Team et al. (2024), and LLaMA-3.3-70B-Instruct-turbo Dubey et al. (2024). For the student model, we choose Mistral-7B-v0.1 Jiang et al. (2023) and Qwen-2.5-14B Yang et al. (2024). To avoid potential preference leakage due to distilling data from other LLMs during the instruction-tuning process, we choose to use the -PRE-TRAINED version rather than the -INSTRUCT version of these student models.

**Evaluation Datasets.** We choose two representative pairwise evaluation datasets, Arena-Hard Li et al. (2024e) and AlpacaEval 2.0 Dubois et al. (2024), to evaluate the trained student models. Arena-Hard includes 500 challenging questions in English. Additionally, the evaluation agreement between Arena-Hard and LMArena Zheng et al. (2023a)'s hard prompts achieved a 96.7% Spearman correlation, demonstrating the consistency of Arena-Hard with human preferences Li et al. (2024e).

AlpacaEval 2.0 is an improved evaluation method based on AlpacaEval Li et al. (2023) and contains 805 questions. Compared to version 1.0, AlpacaEval 2.0 significantly reduces the effect of text length on the evaluation results.

**Implementation Details.** In our main experiment, we examine the preference leakage introduced by using the same data generator and evaluator in supervised fine-tuning (SFT). We will discuss other relatedness and learning methods in Section 5. To obtain synthetic datasets, We first randomly sample 30,000 prompts from the Ultrafeedback dataset Cui et al. (2024). The Ultrafeedback dataset includes instructions from several publicly available high-quality datasets such as TruthfulQA Lin et al. (2022), FalseQA Hu et al. (2023), and Evol-Instruct Xu et al. (2023). For each data generator model, we provide these prompts for them to produce synthetic responses, resulting in three synthetic instruction datasets. We then use each dataset to supervised fine-tune the student model, obtaining three different versions for each baseline: Mistral/ Qwen-GPT-4o, Mistral/ Qwen-Gemini-1.5 and Mistral/ Qwen-LLaMA-3.3. After that, we pair each two student models and obtain three model pairs. For each model pair, we perform the pairwise comparison using the three judge models respectively.

**Metrics** Based on our hypothesis, preference leakage would lead to bias of judge LLMs towards their related student models. Following this principle, we design the preference leakage score $\text{PLS}(i,j)$ to measure the bias in model pair $(i,j)$ caused by preference leakage:

$$\text{PLS}(i,j) = \frac{\left(\frac{\text{WR}(i,i) - \text{AVG}(i,j)}{\text{AVG}(i,j)}\right) + \left(\frac{\text{WR}(j,j) - \text{AVG}(j,i)}{\text{AVG}(j,i)}\right)}{2}, \quad (5)$$

$$\text{AVG}(i,j) = \frac{\text{WR}(i,i) + \text{WR}(i,j)}{2}. \quad (6)$$

Here $\text{WR}(i,j)$ represents the win-rate score from judge model $j$ to student model $i$. Intuitively, a large preference leakage score indicates that the two judge models demonstrate strong bias toward their related student models, suggesting a significant preference leakage phenomenon.

The main experiments in Section 4 and the mitigation analyses in Sections 5.4 and 5.7 are designed for complementary purposes. The main experiments focus on controlled and interpretable measurement of preference leakage itself—quantifying the phenomenon across models and conditions while minimizing confounding factors such as human labeling noise. In contrast, the mitigation analyses prioritize realism and external validity, using human-labeled benchmarks and metrics. Since these setups involve labeled data rather than automatically computed PLS, they serve as realistic extensions that test mitigation feasibility in practical LLM-as-a-judge scenarios. More details about model training and metric explanation can be found in Appendix C.

Table 1: Preference leakage score result on Arena-Hard and AlpacaEval 2.0. The blue background indicates a negative preference leakage score value and the purple background indicates a positive value. The deeper the color, the larger the absolute value.

| Model | Data Generator/ Judge Pair | Arena-Hard | AlpacaEval 2.0 | Avg. |
|---|---|---|---|---|
| | GPT-4o & Gemini-1.5 | 28.7% | 18.4% | 23.6% |
| Mistral-7B | GPT-4o & LLaMA-3.3 | -1.5% | 1.4% | -0.1% |
| | LLaMA-3.3 & Gemini-1.5 | 13.1% | 19.8% | 16.4% |
| | GPT-4o & Gemini-1.5 | 37.1% | 18.6% | 27.9% |
| Qwen-2.5-14B | GPT-4o & LLaMA-3.3 | 1.0% | 2.3% | 1.7% |
| | LLaMA-3.3 & Gemini-1.5 | 25.4% | 18.4% | 21.9% |

## 4.2 MAIN RESULTS

In our main experiment, we aim to provide insights into RQ1.

**Preference leakage exists in most model pairs.** The original judgment results from Arena-Hard and AlpacaEval 2.0, along with the calculated preference leakage scores, are shown in Table 1. As the results demonstrate, in most model pairs (except Mistral-GPT-4o vs Mistral-LLaMA-3.3 and Qwen-GPT-4o vs Qwen-LLaMA-3.3), the judge LLMs exhibit a strong preference toward their related student models, leading to large positive values in the preference leakage scores. This finding

suggests that preference leakage, along with the resulting bias, is widespread in SFT when the data generator and evaluator are the same.

**Smaller student models cause even more bias from judge LLMs.** To investigate the impact of student model size on the degree of preference leakage, we conduct additional experiments using various sizes of the LLaMA-3, Qwen-2.5 and Qwen-3 models. As shown in Figure 4.2 (a), a notable finding is that the smallest models (LLaMA-3-1B, Qwen-2.5-3B and Qwen-3-1.7B) exhibit the highest PL scores than their larger counterparts, indicating greater bias from preference leakage. This trend contrasts with the influence of model size in data contamination, where larger models are typically more susceptible Bordt et al. (2024). We assume that this gap arises from the differing learning capabilities and behaviors of large and small LLMs: while larger models are more prone to memorizing Duan et al. (2024) information that exacerbates data contamination. Compared with them, smaller models may only be able to learn those spurious features that repeatedly occurs (e.g., format), leading to more serious preference leakage.

**Different benchmarks result in varying degrees of bias under preference leakage.** Another observation from Table 1 and Figure 4.2 (a) is that the PL scores in ArenaHard are generally higher than those in AlpacaEval 2.0. One possible explanation is the difference in question difficulty between the two benchmarks, as ArenaHard contains more challenging questions. Additionally, it may also stem from differences in the distribution of question types, the impact of which on preference leakage will be further analyzed in Section 5.6.

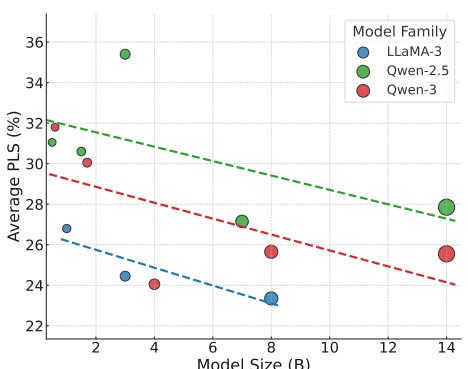

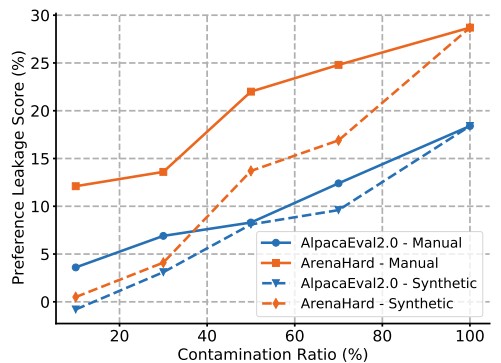

(a) PLS on models with various sizes. We conduct the experiment with GPT-4o and Gemini as data generators and judges.

(b) Experiment results on data mixing. 'Manual' and 'Synthetic represent mixing with manually-written data and other synthetic data, respectively.

Figure 2: Experiment results on additional models and data mixing settings.

# 5 FURTHER ANALYSIS

In this section, we conduct data mixing analysis, relatedness analysis, learning method analysis, and real-world impact analysis (Section 5.1 - 5.4) to answer RQ2. Due to the cost consideration, we conduct these analyses on Mistral-GPT-4o vs Mistral-Gemini-1.5. Moreover, we perform recognition analysis and category analysis to answer RQ3. Additionally, we also benchmark and explore various calibration methods to address preference leakage in Section 5.8.

## 5.1 DATA MIXING ANALYSIS

In real-world applications, synthetic data from a single LLM is often mixed with manually-written data or other multi-source synthetic data to train student models. To mimic these scenarios and explore how much synthetic data could lead to preference leakage, we conduct a data mixing analysis. Specifically, we randomly sample 10%, 30%, 50%, and 70% from the original synthetic dataset and mix it with manually-written data and multi-source synthetic data, respectively, in order to maintain a consistent total volume of training data (30,000). For the manually-written data, we sample from the data pool collected in Section 5.3. For the multi-source synthetic data, we use the original synthetic data from Ultrafeedback, which includes responses generated by various LLMs (e.g., WizardLM, Flcon, etc.). After obtaining the mixing training data, we train the student models using SFT and

calculate their preference leakage scores based on the judgment results. Figure 4.2 (b) presents the results with two mixing strategies across two benchmarks.

**The degree of preference leakage is directly proportional to the amount of synthetic data.** We observe a strong correlation between the proportion of synthetic data in the mixture and the preference leakage score, with no clear threshold separating cases with preference leakage from those without. This suggests that preference leakage can occur even with a small amount of leaked synthetic data, posing significant challenges for its detection.

## 5.2 RELATEDNESS ANALYSIS

We demonstrate the impact of different relatedness conditions between the data generator and the judge LLM on the preference leakage problem, as shown in Table 2.

**Preference leakage under inheritance settings causes obvious bias of judges towards their related students.** For the inheritance relationship, we consider the situation where the data generator is inherited from the judge model. We conducted the following two experiments: (1). we give the same instructions again as in the SFT stage (Inheritance w/ same ins.), or (2). we sample the same number of different instructions from the Ultrafeedback (Inherence w/ different ins.). Then, we let the fine-tuned Mistral model generate the answers and use these generated data to fine-tune a new Mistral student model. From the results, with the same instructions, the average preference leakage score is 19.3%. In comparison, the score with different instructions is 22.3%. Firstly, in an inheritance setting, data generators can inherit judges' preferences, which are then passed on to new student models, thereby compromising the fairness of their evaluation. Second, even when different instructions are used, judges' preferences leaked to data generators can still be transferred to the new student model through synthetic data, leading to a high preference leakage score.

**Models within the same series tend to cause more significant bias.** For two models within the same family, we consider two settings: (1) Same series, where training data is generated by GPT-4o and Gemini-1.5-flash, and judged by GPT-4-turbo and Gemini-1.5-pro; (2) Different series, where training data is still generated by GPT-4o and Gemini-1.5-flash, but judged by GPT-3.5-turbo and Gemini-1.0-pro. In the same series setting, the average preference leakage score is 8.9%, indicating that despite using different models for data generation and judgment,

Table 2: Preference leakage score in different relatedness between the data generator and the judging LLM.

| | Arena-Hard | AlpacaEval 2.0 | Avg. |
|---|---|---|---|
| Same Model | 28.7% | 18.4% | 23.6% |
| Inheritance - w/ same ins. | 17.8% | 20.7% | 19.3% |
| Inheritance - w/ different ins. | 18.3% | 26.3% | 22.3% |
| Same Family - w/ same series | 10.1% | 7.6% | 8.9% |
| Same Family - w/ different series | 3.3% | 2.2% | 2.8% |

their relatedness in terms of model family leads to some preference leakage. In contrast, the different series setting yields a significantly lower leakage score of 2.8%, likely due to differences in architecture, training data, and other factors, reducing the influence of model-related biases in evaluation.

## 5.3 LEARNING METHOD ANALYSIS

We also compare three learning methods, supervised fine-tuning (SFT), direct preference optimization (DPO) Rafailov et al. (2024), and in-context learning (ICL) Dong et al. (2024a), to explore the different influences to them under preference leakage. We first build a data pool based on human-written instruction-tuning data

Table 3: Preference leakage score in different learning methods.

| | Arena-Hard | AlpacaEval 2.0 | Avg. |
|---|---|---|---|
| SFT | 28.7% | 18.4% | 23.6% |
| DPO | 7.7% | 2.7% | 5.2% |
| ICL | -4.2% | -1.1% | -2.7% |

from OASST Köpf et al. (2024), LIMA Zhou et al. (2024), and MOSS Sun et al. (2024b) to supervised fine-tune the pre-trained model. For DPO, we sample 2 responses for each instruction from sampled UltraFeedback instruction and prompt each data generator to produce the pairwise feedback. Then we use the DPO loss to further train the fine-tuned policy on each synthetic pairwise dataset. Appendix E shows the prompt we use to craft synthetic pairwise feedback. For ICL, we sample 4 instruction-response pairs from each LLMs' synthetic dataset as the demonstration during inference.

**Tuning approaches would leak judges' preference to the student models.** Various learning methods show significant differences in preference leakage scores across learning methods. SFT exhibits the highest average leakage score at 23.6%. In contrast, DPO achieves a much lower score of 5.2%, which is consistent with previous studies in data contamination that pairwise optimization can reduce the risk of memorizing or contaminating sensitive training data compared to straightforward supervised fine-tuning Hayes et al.. Meanwhile, ICL, which relies on contextual examples without model tuning, is least affected by the data generator's preferences, resulting in the lowest leakage scores.

## 5.4 REAL-WORLD IMPACT ANALYSIS

Table 4: Impact analysis of preference leakage in real-world LLM-as-a-Judge leaderboards. For each bias type, we assess its impact by calculating the ranking difference of the corresponding model in LMArena and AlpacaEval 2.0, obtained by subtracting the ranking in AlpacaEval 2.0 from that in LMArena. A larger positive ranking difference indicates AlpacaEval 2.0 ranks the target models in higher positions, denoting a greater impact of the corresponding bias.

| Bias Type | Evaluator | Target Models | Ranking Difference |
|---|---|---|---|
| Egocentric Bias | GPT-4 Preview | GPT-4 Preview | 1.00 |
| Preference Leakage | | Vicuna 7B/ 13B/ 33B | **1.33** |

In this section, we investigate the impact of preference leakage in real-world LLM-as-a-Judge leaderboards. While broader leaderboard coverage would enhance external validity, few student–teacher (distillation) pairs are publicly documented, and most leaderboards lack the metadata needed for controlled cross-model comparisons. Moreover, re-evaluating all leaderboard entries with alternate judges would be computationally prohibitive at the current scale. Therefore, we focus on AlpacaEval and LMArena as interpretable case studies and leave large-scale multi-judge re-evaluations for future work. To quantify the effect of each bias type, we calculate the ranking difference of each target model in LMArena and AlpacaEval 2.0.

As shown in Table 4, both egocentric bias and preference leakage result in a positive ranking difference, indicating that both lead to evaluator bias favoring the target models. Notably, the ranking difference associated with preference leakage is even higher than that of egocentric bias, highlighting the substantial impact of preference leakage on real-world LLM-as-a-Judge leaderboards.

## 5.5 CAN JUDGES RECOGNIZE STUDENT MODELS?

Table 5: Student recognition (binary classification) and response classification results (three-class classification). SR: Student Recognition, RC: Response Classification.

| Task | Model | Accuracy | |
|---|---|---|---|
| | | Pointwise | Pairwise |
| SR | GPT-4o | 41.0% | 52.0% |
| | Gemini-1.5 | 53.2% | 44.2% |
| | LLaMA-3.3 | 41.8% | 29.8% |
| RC | BERT | 82.4% | |

Previous studies demonstrate the LLM judges can recognize and thus prefer their own generation Panickssery et al. (2024). In this work, we pose a similar question: *Does preference leakage also source from the LLM judges' recognition of their related student models' generation?* To study this, we follow Panickssery et al. (2024) to prompt the three judge LLMs and test whether they could recognize their related student models' generation. Additionally, we split three student models' generation into training and testing sets, and train a BERT classifier to perform a three-class classification inspired by the previous study on detecting human-AI text Zhang et al. (2024c). For student recognition, we follow Panickssery et al. (2024) to use both pointwise and pairwise settings. Due to the space limitation, more detailed prompting and training settings can be found in Appendix G.

**Judge LLMs do not show good performance in recognizing the generation of their student models.** As the result presented in Table 5, we find that the recognition performance of each judge LLM in the content of related students is poor, with accuracy around the performance of random

guesses. This suggests that preference leakage is subtler and harder-to-detect for judge LLMs, in contrast to the more obvious egocentric bias.

**Certain features embedded in student models through synthetic data.** Although judge LLMs do not perform well in related student recognition, we notice the fine-tuned BERT classification demonstrates a high accuracy score in classifier responses generated by each student model. This suggests that certain characteristics—such as style and format—are embedded in the student models through the synthetic responses. This finding further supports the existence of preference leakage and lays the groundwork for future research aimed at detecting and preventing it. For example, an external detector estimating model-relatedness could provide an auxiliary confidence signal to calibrate or penalize biased judgments.

## 5.6 IMPACT ON QUESTION TYPE & JUDGMENT DIMENSION

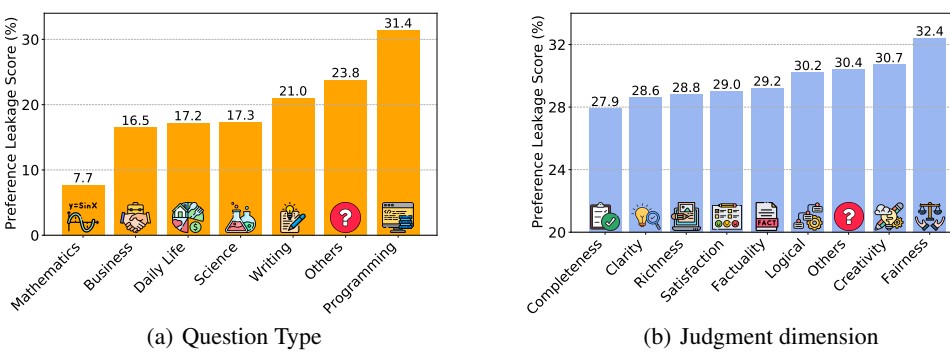

(a) Question Type

(b) Judgment dimension

Figure 3: Category analysis results on question type and judgment dimension.

In this section, we explore the impact of preference leakage across various question types and judgment dimensions. For the question type analysis, we first propose several general question types based on the question clusters introduced by Arena-Hard. Then, we prompt GPT-4o to map each question in Arena-Hard and AlpacaEval to one of the question types and calculate the preference leakage score for each question category. For the judgment dimension analysis, we follow the judgment dimensions introduced by Liu et al. (2023a) and also utilize GPT-4o to map the rationale generated by judge LLMs to one or multiple judgment dimensions. More detailed prompt can be found in Appendix H. The analysis results are presented in Figure 3.

**Subjective question and judgment dimension tend to lead to more bias.** For question type analysis, we find objective questions with a definitive answer, like mathematical ones, demonstrate the least preference leakage. By contrast, subjective questions that have more than one standard answer, such as programming and writing, usually lead to a more obvious preference leakage. This observation is also applied to judgment dimension analysis, as objective dimensions (like completeness) have an overall lower leakage degree compared with subjective ones (like fairness). This suggests that preference leakage tends to be more significant in objective questions and dimensions, where the contaminated model is more likely to receive biased preference.

## 5.7 EFFECT OF SPURIOUS FEATURES ON PREFERENCE LEAKAGE

Table 6: Effect of removing spurious features on the PLS. We consider style, format and wording as potential spurious features in this analysis.

| Setting | GPT & Gemini | GPT & LLaMA | LLaMA & Gemini |
|---------|--------------|-------------|----------------|
| Baseline | 17.5% | 2.3% | 18.8% |
| – w/o style | 9.0% | 3.3% | 14.6% |
| – w/o format | 9.8% | 1.9% | 14.5% |
| – w/o wording | 11.2% | 2.4% | 18.2% |

To further validate that spurious stylistic or formatting cues contribute to preference leakage, we conduct an additional ablation experiment focusing on three major feature categories: style, format,

and wording. Using the Qwen-3-8B model as the rewriting model, we apply a paraphrasing pipeline to selectively remove each type of spurious feature from model responses before evaluation. We employ Gemini-2.0 as the judge model since Gemini-1.5 is no longer available at the time we did this experiment. The rewriting process ensures that the semantic content of each response remains intact while selectively neutralizing surface-level artifacts such as syntactic rhythm, punctuation patterns, and lexical framing cues. By isolating these variables, the experiment provides a more direct lens into how superficial similarity between generator and judge responses shapes preference leakage.

The resulting PLS are reported in Table 6. We observe notable reductions in PLS for the two model pairs that originally exhibited the strongest leakage (GPT & Gemini, LLaMA & Gemini), confirming that removing spurious stylistic alignment substantially mitigates bias. Among the three feature types, eliminating style and format yields the largest decrease in leakage, suggesting that judges tend to rely heavily on stylistic regularities—such as tone consistency, sentence cadence, and punctuation density—when forming preference judgments. In contrast, removing wording-level features (e.g., synonym substitution or phrase order changes) produces only minor improvements, implying that lexical similarity alone is not the dominant driver. Interestingly, the magnitude of reduction varies across judge families: GPT-based judges appear especially responsive to stylistic coherence, while LLaMA-based judges are more influenced by formatting regularity. This diversity in sensitivity indicates that each model family has distinct perceptual priors about linguistic structure, which can amplify different forms of spurious correlation. Overall, these findings empirically substantiate our mechanistic explanation that stylistic and formatting artifacts embedded in student models act as hidden conduits for preference leakage, shaping judge behavior through subtle surface-level mimicry rather than semantic alignment.

## 5.8 EXPLORING MITIGATION METHOD FOR PREFERENCE LEAKAGE

To benchmark and explore mitigation methods for preference leakage, we collected human-labeled pairwise judgments from several reward benchmarks, including PPE Perez et al. (2022), MTBench Zheng et al. (2023b), and Human Preference Chiang et al. (2024). Using GPT-4 as the target model, we selected samples in which one of the responses was generated by GPT-4's related student (e.g., Vicuna, Alpaca). We then tested several mitigation methods on this dataset, including prompting, chain-of-thought

Table 7: Error Bias with various mitigation methods (lower is better).

| Method | Error Bias |
|---|---|
| Base | 17.8 |
| + Prompting | 18.3 |
| + Chain-of-Thought | 15.6 |
| + Paraphrase | 18.7 |
| + Auto Calibration | 20.7 |
| + Contextual Calibration | **7.3** |

(CoT), paraphrasing, auto-calibration, and contextual calibration. The explored mitigation strategies can be grouped into two complementary layers: (i) Input- or reasoning-level debiasing (prompting, CoT, paraphrasing) that modifies inputs or reasoning chains; and (ii) Output-level calibration (auto- or contextual calibration) that adjusts scores post-hoc. We further propose a new metric, Error Bias, based on human-labeled judgments: $\text{ErrorBias} = \frac{N_{\text{target-prefer-other-win}}}{N_{\text{other-win}}} - \frac{N_{\text{other-prefer-target-win}}}{N_{\text{target-win}}}$. Intuitively, this metric quantifies the difference between target-preferred errors and other-preferred errors; a value close to 0 indicates that preference leakage is mitigated. Our preliminary results show that contextual calibration with an additional held-out set for bias adjustment is the most effective, reducing Error Bias from 17.8 to 7.3. We provide a more detailed explanation about each method in Appendix C.4.

## 6 CONCLUSION

In this work, we formally highlight the preference leakage problem in LLM-as-a-judge systems. The results of our main experiment, measured using the proposed preference leakage score, reveal a clear bias in each judge toward their respective student model. We also observe that this bias is more pronounced in certain question types and smaller student models. Furthermore, we conduct additional analysis on various factors, including the relationship between the data generator and judge LLMs, model tuning techniques, data mixing strategies, and real-world applications. Our findings suggest that preference leakage can cause significant bias across diverse scenarios. Finally, we investigate the underlying mechanisms of preference leakage, demonstrating that it is a challenging and hard-to-detect issue, especially in subjective questions and judgment dimensions.

## ETHICS STATEMENT

We adhere to the ICLR Code of Ethics. No private, sensitive, or personally identifiable data are involved. Our work does not raise foreseeable ethical concerns or produce harmful societal outcomes.

## REPRODUCIBILITY STATEMENT

Reproducibility is central to our work. All datasets used in our experiments are standard benchmarks that are publicly available. We provide full details of the training setup, model architectures, and evaluation metrics in the main paper and appendix. Upon acceptance, we will release our codebase, including scripts for preprocessing, training, and evaluation, along with configuration files and documentation to facilitate exact reproduction of our results. Random seeds and hyperparameters will also be included to further ensure reproducibility.

## ACKNOWLEDGEMENT

This work was supported by the U.S. Department of Defense, Army Research Office (Award No. W911NF-24-2-0175). The views and conclusions contained in this document are those of the authors and should not be interpreted as representing official policies, either expressed or implied, of DOD or ARO.

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

## A    THE USE OF LLMS FOR WRITING

We employed Google's Gemini 2.5 Pro and OpenAI's GPT-5 as writing assistance tools during the preparation of this manuscript. Their role was exclusively for language refinement, such as improving readability and rephrasing for clarity in an academic writing style. This usage aligns with standard academic practices for language polishing.

## B    PRELIMINARY STUDY OF PREFERENCE LEAKAGE IN REAL WORLD

In our preliminary study, we investigate whether preference leakage is a real-world issue in mainstream leaderboards and benchmarks. To this end, we examine two widely used LLM-as-a-judge leaderboards (AlpacaEval 2.0 and Arena-Hard) and a well-known benchmark (MTBench). All three rely on GPT-4 as the judge model and report pairwise judgment results for various LLMs. Our analysis reveals that several candidate models distilled from GPT-4 or other GPT-series models (e.g., Vicuna and Alpaca) appear across all these leaderboards and benchmarks, suggesting that preference leakage is a pervasive issue in these datasets. Besides, we also examine if preference leakage exists in LLM-relevant research studies and also find a bunch of work utilizing the same or related model(s) to do distillation/ data synthesis and evaluation Yang et al. (2023); Liu et al. (2024a); Lee et al. (2024); Li et al. (2024d); Wang et al. (2024); Sun et al. (2024a). All of these suggest preference leakage to be a widespread problem in both LLM-as-a-judge datasets and LLM-relevant research.

## C    EXPERIMENT DETAILS

### C.1    TRAINING DETAILS

We use LLaMA-Factory Zheng et al. (2024), an efficient LLM tuning library for our experiment. The maximum sequence length is set to 1024 tokens, and a cutoff length of 1024 tokens is enforced to prevent excessive tokenization. The data preprocessing will be done in parallel with 16 workers to speed up the preparation process. The training use a per-device batch size of 2, with gradient accumulation over 2 steps to simulate a larger batch size for SFT and a per-device batch size of 1, with gradient accumulation over 4 steps to simulate a larger batch size for DPO. The learning rate is set to 1.0e-5 and each model will be trained for 3 epochs. A cosine learning rate scheduler is used with a warmup ratio of 0.1 to gradually increase the learning rate during the initial steps. All of the experiments use BF16 precision to speed up training while maintaining numerical stability. All the experiments are conducted in an 8 Nvidia A100 GPU cluster with CUDA version 11.8.

Table 8: A case on AlpacaEval 2.0 with the model pair Mistral-GPT-4o vs Mistral-Gemini-1.5 to demonstrate how the preference leakage score is calculated.

| Judge Model | Mistral-GPT-4o vs Mistral-Gemini-1.5 | |
|---|---|---|
| | Mistral-GPT-4o Wins | Mistral-Gemini-1.5 Wins |
| GPT-4o | 55.1% | 44.9% |
| Gemini-1.5 | 36.8% | 63.2% |
| Preference Leakage Score | 18.4% | |

### C.2    DETAILED EXPLANATION FOR PREFERENCE LEAKAGE SCORE

We present a case in Table 8 to show how we calculate the preference leakage score for the Mistral-GPT-4o vs Mistral-Gemini-1.5 pair on AlpacaEval 2.0. Based on the definition of preference leakage score, we first calculate:

$$\text{AVG}(\text{Mistral-GPT-4o}, \text{Mistral-Gemini-1.5}) = \frac{55.1 + 36.8}{2} = 45.95\% \qquad (7)$$

$$\text{AVG}(\text{Mistral-Gemini-1.5}, \text{Mistral-GPT-4o}) = \frac{63.2 + 44.9}{2} = 54.05\% \qquad (8)$$

After that, we calculate the preference leakage score:

$$\text{PLS}(\text{Mistral-GPT-4o}, \text{Mistral-Gemini-1.5}) = \frac{\left(\frac{55.1-45.95}{45.95}\right) + \left(\frac{63.2-54.05}{54.05}\right)}{2} = 18.4\% \quad (9)$$

.

### C.3 Manual Annotation Details & Results

While we have concluded that student model pairs with similar performance or more powerful student models tend to exhibit greater preference leakage, we also examine whether different data generator and judge LLMs contribute to varying degrees of preference leakage. We randomly sample 100 questions from AlpacaEval 2.0 and ask three well-trained annotators to conduct pairwise comparisons of the responses from each model pair for these questions. For annotation efficiency, we also develop an annotation tool that involves the function of uploading multiple model responses, jumping to specific problems, and downloading annotation results (Figure 7). After annotation, we adopt the majority voting to get the final label for each response pair. We also calculate the average agreement of three annotators and find it to be 78.6, indicating a relatively consistent annotation result.

Analyzing the manual annotation results presented in Figure 4, we observe that Gemini-1.5 shows a strong bias toward its students, followed by GPT-4o, with LLaMA-3.3 displaying the least bias. This variation in preference leakage may stem from differences in the level of leaked preference in the synthetic responses generated by the data generator LLMs. For instance, an LLM with a distinctive style or format in its responses offers more opportunities for student models to learn these characteristics, potentially leading to more pronounced preference leakage during evaluation. Future work could further quantify the extent of leaked preference for each data generator model.

### C.4 Mitigation Methods Details

**Dataset Construction.** To systematically benchmark preference leakage, we curate a pairwise judgment corpus by consolidating three widely used human–labeled reward datasets: **PPE** Perez et al. (2022), **MTBench** Zheng et al. (2023b), and the **Human Preference** dataset Chiang et al. (2024). Each dataset contains prompts and paired model outputs annotated with human preferences. We treat GPT-4 as the *target model* and identify instances where one response originates from GPT-4 and the other from a related open–source "student" model (e.g., Vicuna, Alpaca).

**Mitigation Methods.** We evaluate five representative strategies designed to counteract preference leakage:

- **Prompting.** A straightforward baseline that refines evaluation instructions to explicitly warn against self-preference, encouraging the evaluator to remain impartial and judge outputs solely on content quality and relevance.

- **Chain-of-Thought (CoT).** Augments the evaluation prompt by encouraging the model to articulate an explicit step-by-step reasoning process prior to producing its final decision, thereby reducing unconscious style matching.

- **Paraphrasing.** Reduces lexical and stylistic overlap between the evaluator and candidate outputs by paraphrasing prompts or responses before evaluation, mitigating familiarity-driven bias.

- **Auto-Calibration.** Estimates a global bias term from a held-out calibration set by analyzing the evaluator's log-probabilities of choosing the target versus the student, then shifts future predictions to offset this bias.

- **Contextual Calibration.** Extends auto-calibration by learning context-dependent bias adjustments. For each evaluation scenario, bias is estimated from a similar held-out set and applied dynamically at inference time, offering finer-grained debiasing and achieving the strongest reduction in preference leakage.

## D Additional Experiments

Due to the space limitation, we put further experiments and analysis in the Appendix.

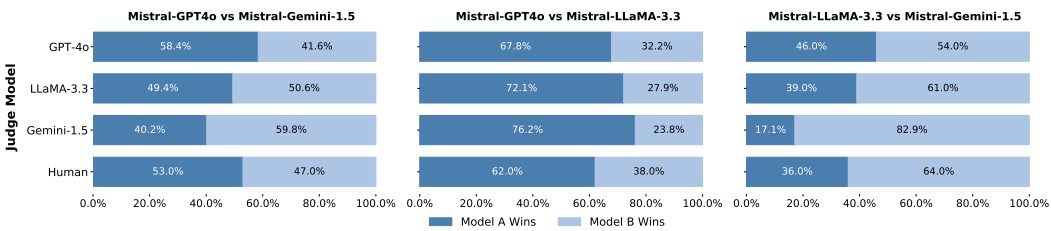

Figure 4: Manual annotation result on 100 randomly selected samples from AlpacaEval 2.0.

## D.1 ORIGINAL EXPERIMENT RESULTS FOR PLS CALCULATION

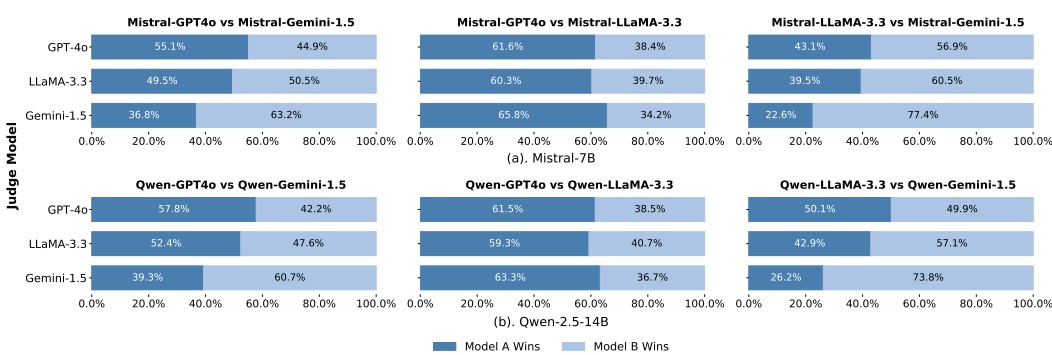

Figure 5: Judgment results with GPT-4o, LLaMA-3.3 and Gemini-1.5 on AlpacaEval 2.0. Different from Arena-Hard, there is no tie in AlpacaEval 2.0.

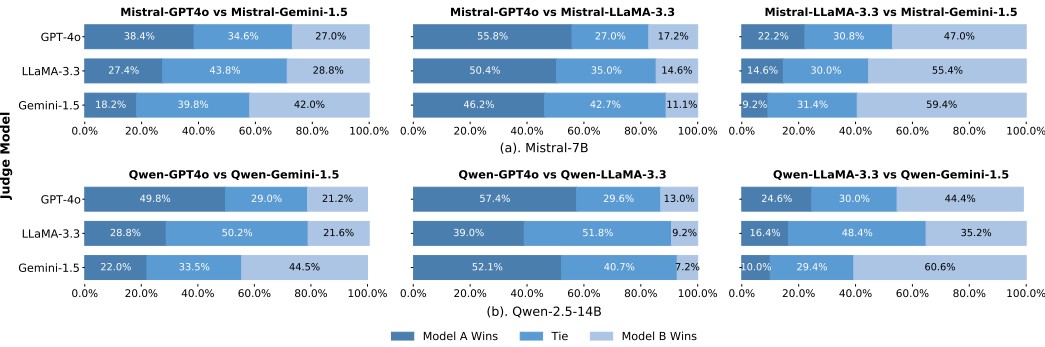

Figure 6: Judgment results with GPT-4o, LLaMA-3.3 and Gemini-1.5 on Arena-Hard.

## D.2 STABILITY ASSESSMENT OF EXPERIMENTAL RESULTS

Based on the results from three repeated experiments (Table 9), we observe consistently low variance across different comparisons, indicating high stability in performance measurements. This suggests that the conclusions drawn from these experiments are reliable and not significantly affected by random fluctuations, thereby strengthening the validity of our findings.

## D.3 PROMPT SENSITIVITY ANALYSIS

We examined the robustness of the *Preference Leakage Score (PLS)* under different evaluation prompts. Two LLM-as-a-judge protocols were used: ARENAHARD and ALPACAEVAL 2.0, each with distinct prompts and question sets. We rewrote the prompts for both protocols and re-ran the evaluations.

Table 9: Mean and variance of experimental results across two benchmarks in Mistral-7B-v0.1.

| Model Pairs | Mean | Variance |
|---|---|---|
| *ArenaHard* | | |
| mistral-GPT4o vs mistral-Gemini-3.3 | 28.67 | 0.063 |
| mistral-GPT4o vs mistral-LLAMA-3.3 | 0.50 | 0.910 |
| mistral-LLAMA vs mistral-Gemini | 12.93 | 0.583 |
| *AlpacaEval 2.0* | | |
| mistral-GPT4o vs mistral-Gemini-3.3 | 19.20 | 0.490 |
| mistral-GPT4o vs mistral-LLAMA-3.3 | 0.20 | 1.240 |
| mistral-LLAMA vs mistral-Gemini | 19.87 | 0.013 |

Table 10: PLS under different evaluation prompts.

| Judge Pair | Prompt 1 | Prompt 2 | Dataset |
|---|---|---|---|
| GPT-4o vs Gemini-1.5 | 18.4% | 16.5% | AlpacaEval 2.0 |
| | 28.7% | 38.7% | ArenaHard |
| GPT-4o vs LLaMA-3.3 | 1.4% | -1.2% | AlpacaEval 2.0 |
| | -1.5% | 4.5% | ArenaHard |
| LLaMA-3.3 vs Gemini-1.5 | 19.8% | 17.9% | AlpacaEval 2.0 |
| | 13.1% | 15.8% | ArenaHard |

PLS remained consistently $> 0$ for key model pairs; ALPACAEVAL 2.0 was more stable to prompt changes than ARENAHARD.

## D.4 STATISTICAL SIGNIFICANCE TESTS

We tested the hypothesis PLS $> 0$ using a non-parametric bootstrap with 10,000 resamples over 500 prompts in ARENAHARD.

Table 11: Bootstrap significance results for PLS $> 0$. ***: $p < 0.001$, **: $p < 0.01$.

| Judge Pair | Student | PLS (%) | Significance |
|---|---|---|---|
| GPT-4o vs Gemini-1.5 | Mistral-7B | 28.5 | *** |
| GPT-4o vs LLaMA-3.3 | Mistral-7B | -1.1 | n.s. |
| LLaMA-3.3 vs Gemini-1.5 | Mistral-7B | 7.4 | ** |
| GPT-4o vs Gemini-1.5 | Qwen-2.5-14B | 37.9 | *** |
| GPT-4o vs LLaMA-3.3 | Qwen-2.5-14B | 1.2 | n.s. |
| LLaMA-3.3 vs Gemini-1.5 | Qwen-2.5-14B | 26.3 | *** |

## D.5 LANGUAGE GENERALIZATION

To test cross-lingual generalization, we synthesized Chinese SFT data (using Moss-3 instructions) and evaluated with Chinese versions of ARENAHARD (m-ARENAHARD) and XALPACAEVAL. Judges were GPT-4o and Gemini-1.5; the student model was Qwen-3-8B.

Significant preference leakage also appears in the Chinese setting.

## D.6 EXPANDED JUDGE–STUDENT PAIRS

We added the judge model Claude-3.5-Sonnet to form three new judge pairs: GPT-4o & Claude-3.5, Gemini & Claude-3.5, and LLaMA-3.3 & Claude-3.5. Student models: Mistral-7B and Qwen-2.5-14B.

Table 12: PLS in English vs. Chinese.

| Language | AlpacaEval 2.0 | ArenaHard | Avg |
|---|---|---|---|
| English | 17.4% | 33.9% | 25.7% |
| Chinese | 12.3% | 51.8% | 32.1% |

Table 13: PLS of new judge pairs (negative values indicate no leakage).

| Judge Pair | ArenaHard | AlpacaEval 2.0 | Avg |
|---|---|---|---|
| **Mistral-7B** | | | |
| GPT-4o & Claude-3.5 | 12.2% | 8.6% | 10.4% |
| Gemini & Claude-3.5 | 16.5% | 7.1% | 11.8% |
| LLaMA-3.3 & Claude-3.5 | -4.4% | -2.6% | -3.5% |
| **Qwen-2.5-14B** | | | |
| GPT-4o & Claude-3.5 | 13.0% | 10.4% | 11.7% |
| Gemini & Claude-3.5 | 18.5% | 11.1% | 14.8% |
| LLaMA-3.3 & Claude-3.5 | 0.0% | 1.7% | 0.9% |

## D.7 STUDENT MODEL SCALING

We tested PLS on a wider range of student sizes within the Qwen and LLaMA families.

Table 14: PLS (%) for different student sizes.

| Student | ArenaHard | AlpacaEval 2.0 | Avg |
|---|---|---|---|
| LLaMA-3-1B | 35.4 | 18.2 | 26.8 |
| LLaMA-3-3B | 32.5 | 16.4 | 24.5 |
| LLaMA-3-8B | 30.9 | 15.8 | 23.4 |
| Qwen-2.5-0.5B | 40.9 | 21.2 | 31.1 |
| Qwen-2.5-1.5B | 38.0 | 23.2 | 30.6 |
| Qwen-2.5-3B | 50.7 | 20.1 | 35.4 |
| Qwen-2.5-7B | 32.2 | 22.1 | 27.2 |
| Qwen-2.5-14B | 37.1 | 18.6 | 27.9 |
| Qwen-3-0.6B | 39.8 | 23.8 | 31.8 |
| Qwen-3-1.7B | 40.0 | 20.1 | 30.2 |
| Qwen-3-4B | 30.9 | 17.2 | 24.1 |
| Qwen-3-8B | 33.9 | 17.4 | 25.7 |
| Qwen-3-14B | 31.7 | 19.4 | 25.6 |

Within each family, smaller models generally exhibit higher PLS.

## D.8 MITIGATION METHODS AND ERROR BIAS METRIC

We explored mitigation methods on a human-labeled reward dataset, including: prompting, chain-of-thought (CoT), paraphrasing, auto-calibration, and contextual calibration. We introduced the *Error Bias* metric:

$$\text{ErrorBias} = \frac{N_{\text{target-prefer-other-win}}}{N_{\text{other-win}}} - \frac{N_{\text{other-prefer-target-win}}}{N_{\text{target-win}}}. \tag{10}$$

Contextual calibration with an additional held-out bias-adjustment set yielded the largest reduction.

## E LEARNING METHOD ANALYSIS DETAILS

The table below presents the prompt we use to generate synthetic pairwise feedback from each model.

---

**Pairwise Feedback Prompt**

```
Please act as an impartial judge and evaluate the quality of the
responses provided by two AI assistants to the user question
displayed below.  Your evaluation should consider correctness
and helpfulness.  You will be given assistant A's answer, and
assistant B's answer.  Your job is to evaluate which assistant's
answer is better.  You should independently solve the user question
step-by-step first.  Then compare both assistants' answers with
your answer.  Identify and correct any mistakes.  Avoid any
position biases and ensure that the order in which the responses
were presented does not influence your decision.  Do not allow
the length of the responses to influence your evaluation.  Do not
favor certain names of the assistants.  Be as objective as possible.
After providing your explanation, output your final verdict by
strictly following this format:  "[[A]]" if assistant A is better,
"[[B]]" if assistant B is better.

## Instruction:

[The Start of Assistant A's Answer]
[RESPONSE A]
[The End of Assistant A's Answer]

[The Start of Assistant B's Answer]
[RESPONSE B]
[The End of Assistant B's Answer]

Please output the generated content in a json format, for example:
{ "reason":  // string, reasons behind the chosen preferred answer
"prefered answer":  // string, the prefered answer you selected,
[[A]] or [[B]] }

Formatted the abovementioned schema and produce the reason and
preferred answer:
```

---

## F   REAL-WORLD IMPACT ANALYSIS DETAILS

In the real-world impact analysis section, we use the models that appear in both LMArena and AlpacaEval 2.0 leaderboard, including: GPT-4o-2024-05-13, GPT-4o-mini-2024-07-18, Meta-Llama-3.1-405B-Instruct-bf16, GPT-4-Turbo-2024-04-09, GPT-4-1106-preview, Meta-Llama-3.1-70B-Instruct, Claude 3 Opus, Llama-3-70B-Instruct, Claude 3 Sonnet, Qwen2-72B-Instruct, GPT-4-0314, Meta-Llama-3.1-8B-Instruct, GPT-4-0613, Mistral-Large-2402, Llama-3-8B-Instruct, Command R (04-2024), Mistral Medium, Mixtral-8x22b-Instruct-v0.1, Qwen1.5-72B-Chat, Gemini Pro, Yi-34B-Chat, Mixtral-8x7B-Instruct-v0.1, Qwen1.5-14B-Chat, GPT-3.5-Turbo-0125, DBRX-Instruct-Preview, Tulu-2-DPO-70B, Llama-2-70B-chat, Vicuna-33B, Gemma-1.1-7B-it, OpenHermes-2.5-Mistral-7B, Mistral-7B-Instruct-v0.2, Qwen1.5-7B-Chat, GPT-3.5-Turbo-1106, Llama-2-13b-chat, WizardLM-13b-v1.2, Vicuna-13B, Llama-2-7b-chat, Guanaco-33B, Vicuna-7B, Gemma-2B-it, OpenAssistant-Pythia-12B.

## G   RECOGNITON ANALYSIS DETAILS

The table below presents the pointwise and pairwise prompts we use for the recognition analysis.

**Pointwise Recognition Prompt**

```
Given an instruction and a response, your task is to judge
whether this response is generated by a model that is trained on
a synthetic dataset you produced (your student model).
```

```
## Instruction:  [INSTRUCTION]

## Response:  [Response]

Please output the generated content in a json format, for example:
"reason":  // string, reasons behind the judgment "judgment":  //
string, whether the answer is generated by your student model,
choose from yes or no

Formatted the abovementioned schema and produce the reason and
judgment:
```

**Pairwise Recognition Prompt**

```
Given an instruction and two responses, your task is to judge which
response is generated by a model that is trained on a synthetic
dataset you produced (your student model).

## Instruction:  [INSTRUCTION]

## Response1:  [Response 1]

## Response2:  [Response 2]

Please output the generated content in a json format, for example:
"reason":  // string, reasons behind the judgment "judgment":  //
int, 1 or 2, means response1 or response2 is from your student
model

Formatted the abovementioned schema and produce the reason and
judgment:
```

For response classification, we split all the response from three student models into training (80%) and testing (20%) subsets. Then, we finetune a BERT-base-uncased model in the training set. The model is trained for 3 epochs with a learning rate of 2e-5, a batch size of 16 for both training and evaluation, and a weight decay of 0.01, with evaluations conducted at the end of each epoch.

## H  CATEGORY ANALYSIS DETAILS

The tables below present the prompt we use for question type and judgment dimension cateogory analysis.

**Question Type Categorization Prompt**

```
Given a question, please categorize it to one of the following
categories:

1.  Computer Science & Programming
2.  Mathematics & Statistics
3.  Science & Engineering
4.  Business & Finance
5.  Writing & Communication
6.  Social & Daily Life
7.  Others

## Question:  [QUESTION]
```

```
Please output the generated content in a json format, for example:
{ "question category":  // string, specific category name, such as
"Computer Science & Programming" }

Formatted the abovementioned schema and categorize the given
question:
```

**Judgment Dimension Categorization Prompt**

```
Given a pairwise comparison judgment made by an AI, please
categorize each considered aspect in the rationale to one of the
following categories:

{

"Factuality":  "Whether the information provided in the response is
accurate, based on reliable facts and data.",

"User Satisfaction":  "Whether the response meets the user's
question and needs, and provides a comprehensive and appropriate
answer to the question.",

"Logical Coherence":  "Whether the response maintains overall
consistency and logical coherence between different sections,
avoiding self-contradiction.",

"Richness":  "Whether the response includes rich info, depth,
context, diversity, detailed explanations and examples to meet user
needs and provide a comprehensive understanding.",

"Creativity":  "Whether the response is innovative or unique,
providing novel insights or solutions.",

"Fairness and Responsibility":  "Whether the advice or information
provided in the response is feasible, carries acertain degree of
responsibility, and considers potential risks and consequences.",

"Completeness":  "Whether the response provides sufficient
information and details to meet the user's needs, and whether it
avoids omitting important aspects.",

"Clarity":  "Whether the response is clear and understandable, and
whether it uses concise language and structure so that the user can
easily understand it.",

"Others":  "Other aspects which is not listed above."
}

## Judgment:  [JUDGMENT]

Please output the generated content in a json format, for example:
{ "Factuality":  // list, all aspects that belong to this category,
such as ["correctness", "mistakes"] ...  }

Formatted the abovementioned schema and categorize aspects in the
judgment:
```

## I  BROADER IMPACT

By revealing preference leakage, this work could help build more trustworthy and ethically grounded AI systems. The relatedness between data generators and evaluators can systematically bias evaluations, potentially compromising the fairness and reliability of the automatic evaluation paradigm. These biased evaluations may indirectly affect downstream tasks such as AI alignment and decision-making systems, leading to unintended ethical risks. To mitigate preference leakage, we hope that

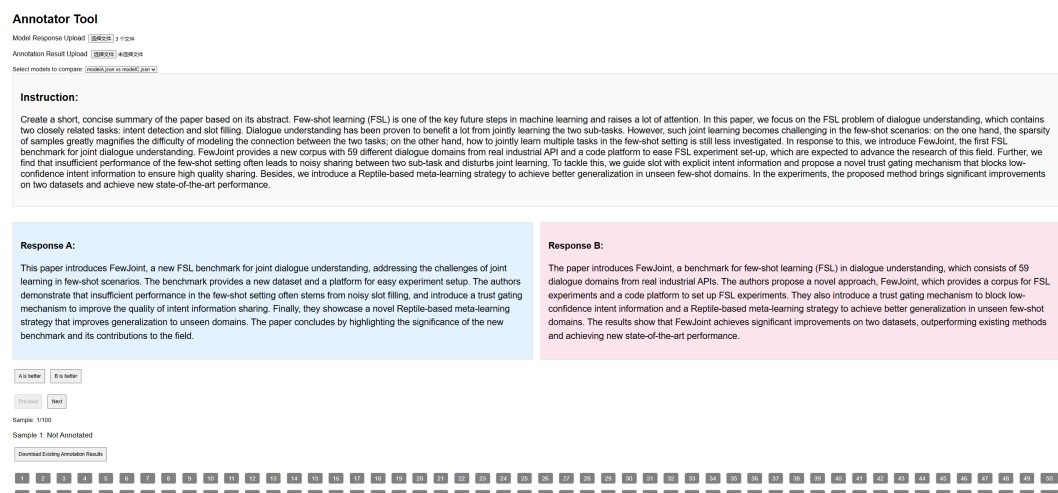

Figure 7: The annotation tool we develop for annotation efficiency.

researchers will propose more reliable evaluation methods, diversify training data sources, and develop contamination-resistant benchmarks in the future.

