# OpenReview forum: "Preference Leakage: A Contamination Problem in LLM-as-a-judge"
_ICLR.cc/2026/Conference — ICLR 2026 Poster_

### Official Review · Reviewer_fJqb · 2025-10-28

**Soundness:** 3
**Presentation:** 3
**Contribution:** 3
**Rating:** 6
**Confidence:** 3

**Summary:**

The paper proposes preference leakage, a contamination problem in LLM-as-a-judge. The paper identifies three common types of relatedness between data generator LLM and the judge LLM, and confirms that preference leakage causes LLM judge to prefer related student models via comprehensive experiments. In addition, LLM judges are not good at identifying responses from student models, indicating that preference leakage is hard to detect.

**Strengths:**

The paper proposes a novel concept of contamination in the domain of LLM-as-a-judge, and identifies the preference bias caused by such contamination. The paper conducts sufficient experiments to support its claim, raising new questions to LLM-as-a-judge research community on how to prevent or identify preference leakage.

**Weaknesses:**

The paper does not study in detail how to detect preference leakage, or how to mitigate preference leakage. Although they provide preliminary analyses in Section 5.7, the methods explored are general and not designed specifically for preference leakage mitigation. Considering that preference leakage is a novel concept, these weaknesses are acceptable.

**Questions:**

Is there any effective method to detect preference leakage, especially the inheritance relationship? For instance, would it help to measure the correlation coefficient of the last hidden state between judge and student on the same prompt?

---

> ### Author Response · Authors · 2025-11-22
>
> We sincerely thank the reviewer for the acknowledgement of our work. We would like to provide further clarification and explanation for the following points.
>
> **Mitigation Methods.** As we state in Section 5.7, our mitigation experiments intentionally adapt several general debiasing and calibration techniques (prompting, CoT, paraphrasing, auto-/contextual calibration) to the preference-leakage (PL) setting; these choices were practical first-steps intended to establish baselines and show that the PL signal can be meaningfully reduced in realistic, labeled benchmarks. Because this paper’s core contribution is the identification, quantification and empirical analysis of preference leakage (Sections 3–5), we prioritized breadth of diagnosis over an exhaustive search for bespoke PL-specific countermeasures. Space constraints therefore limited how deeply we could pursue PL-tailored mitigation designs in this submission. We agree that PL-specific techniques are a crucial next step and in the follow-on work we will do more exploration.
>
> **Detecting Methods.** Detecting PL (and especially inheritance relationships) is indeed a central open problem and we welcome the reviewer’s suggested direction. Correlating representation vectors (e.g., last hidden states) between judge and student on the same prompts is a promising and intuitive signal: high representational similarity on many inputs could indicate that the student has internalized generator preferences and thus is at risk of being preferred by a related judge. In Section 5.5 we already show that a supervised classifier (BERT) can reliably distinguish student generations even when judges themselves cannot — this result demonstrates that representation- or classifier-based detection is feasible and provides a natural starting point for more automated detectors. We also note that several contemporaneous works on “distillation detection” are beginning to appear [1][2], underscoring that this is an emergent, open-ended challenge.
>
> We hope our response addresses the reviewer’s concerns and will be open to any further discussion!
>
> [1] Wadhwa, Somin, et al. "Who taught you that? tracing teachers in model distillation." arXiv preprint arXiv:2502.06659 (2025).
> [2] Li, Pingzhi, et al. "Leave It to the Experts: Detecting Knowledge Distillation via MoE Expert Signatures." arXiv preprint arXiv:2510.16968 (2025).

---

> > ### Comment · Reviewer_fJqb · 2025-11-25
> >
> > Thanks for the detailed response from the authors. It resolves my concern about mitigation and detection methods. Though the work does not propose novel mitigation approaches of preference leakage, it dives deeply enough to the identification issue. Therefore, I believe it makes enough contribution to be a good paper. I've raised my score to 8.

---

> > > ### Author Response · Authors · 2025-11-26
> > > **Thanks for reconsidering our work**
> > >
> > > We really appreciate the reviewer for reconsidering our work. The raised points, including mitigation and detection methods for PL, are informative and thoughtful. We will definitely explore more approaches to address and mitigate PL in the future.

---

### Official Review · Reviewer_HBz8 · 2025-10-28

**Soundness:** 4
**Presentation:** 4
**Contribution:** 3
**Rating:** 8
**Confidence:** 5

**Summary:**

The paper investigates preference leakage, a newly defined contamination phenomenon in LLM-as-a-Judge frameworks, where bias arises when data-generation and evaluation models are related (same model, inheritance, or same model family). Through large-scale experiments across GPT-4o, Gemini, and LLaMA-3 series, the authors show that related generator–judge pairs inflate evaluation scores, compromising benchmark reliability. They design a Preference Leakage Score (PLS) and analyze the issue under multiple conditions (data mixing, model size, learning method, real-world leaderboards, etc.), concluding that preference leakage is widespread and subtle compared to data leakage or egocentric bias.

**Strengths:**

1 Novel and important concept: The notion of preference leakage extends beyond existing bias categories and raises an underexplored but crucial reliability issue in LLM evaluation.

2 Systematic empirical validation: Comprehensive experiments using diverse generator-judge relations, datasets (Arena-Hard, AlpacaEval 2.0), and mitigation trials.

3 Strong methodological framework: Clear definitions of “relatedness” and formalization of leakage conditions make this paper theoretically grounded.

4 Practical relevance: Direct implications for the credibility of public leaderboards and synthetic-data pipelines. It will inspire industry focus on this area of preference leakage.

5: The Preference Leakage Score (PLS)  is innovative and worth to further explore.

**Weaknesses:**

1 The discussion of features embedded in student models (Sec 5.5) is promising but under-analyzed; deeper probing or visualization (e.g., stylistic feature attribution) would enrich understanding.

2 Some family coverage is narrow—adding DeepSeek and Grok series could strengthen generality and industry relevance.

3 If “preference” is positioned as a form of affective bias, connections to affective-analysis or sentiment-evaluation benchmarks would contextualize it better.

4 While mitigation methods (Sec 5.7) are valuable, quantitative comparisons among them are brief; more ablation or human-alignment results would improve credibility.

**Questions:**

1 In Section 3.3, how do the authors define “same model family” when architectures diverge but pre-training corpora overlap (e.g., Mistral vs Gemma)?

2 Can the authors provide significance testing or confidence intervals for the Preference Leakage Score (PLS) results in Table 1 and Figure 2?

3 In the learning method analysis (Section 5.3), how do the authors distinguish the leakage reduction effects of DPO from general overfitting control—could this be further verified through ablation?

4 What specific linguistic or structural features appear to be “embedded” in student models through synthetic data (Section 5.5)? Are they stylistic, lexical, or formatting-based?

5 Could contextual calibration (Section 5.7) be combined with adversarial or paraphrased prompting to further mitigate preference leakage?

6 Given that preference leakage may relate to model “affective preferences,” have the authors considered testing the phenomenon on affective or sentiment-analysis benchmarks?

7 To what extent might cross-family transfer (e.g., DeepSeek ↔ Grok ↔ Gemini) reveal broader generalization or hidden correlations beyond single-family leakage?

---

> ### Author Response · Authors · 2025-11-22
> **Response (part-1)**
>
> We sincerely thank the reviewer for the acknowledgement of our work. We would like to provide further clarification and explanation for the following points.
>
> **Effect of Spurious Feature on Preference Leakage.** We thank the reviewer for this insightful comment. To further validate that spurious features (style, format, wording) contribute to preference leakage, we conducted an additional ablation experiment. Using Qwen-3-8B as the rewriting model, we applied a paraphrasing pipeline to selectively remove these features from model responses before evaluation. Because Gemini-1.5 is no longer available, we used Gemini-2.0 as the judge model. The table below shows the new PLS scores after removing each feature type. We observe significant PLS reductions in the two pairs that initially showed strong leakage (GPT & Gemini, LLaMA & Gemini), confirming that spurious features are key contributors. Among the three features, style and format removal produce the largest decrease, whereas wording has a smaller effect. Differences across model pairs suggest that distinct judges are sensitive to different spurious traits—e.g., GPT-based judges may favor stylistic similarity more than LLaMA-based ones. These findings empirically substantiate our mechanistic explanation of preference leakage. We appreciate the reviewer’s encouragement to deepen the analysis. The above ablation directly supports our earlier hypothesis that spurious stylistic and formatting patterns embedded in student models drive biased judgments. We will include this experiment and its detailed implementation (prompting design, paraphrasing strategy, and model setup) in the revision.
>
> | Setting | GPT & Gemini | GPT & LLaMA | LLaMA & Gemini |
> |----------|---------------|--------------|----------------|
> | Baseline | 17.5% | 2.3% | 18.8% |
> | – w/o style | 9.0% | 3.3% | 14.6% |
> | – w/o format | 9.8% | 1.9% | 14.5% |
> | – w/o wording | 11.2% | 2.4% | 18.2% |
>
>
> **On the Definition of “same model family”.** We appreciate the reviewer’s question. In our paper, “same model family” is defined primarily by organizational provenance — that is, models released by the same LLM provider or research organization. This reflects a realistic industrial setting in which the same team’s data generator and evaluator are often entangled in the model development pipeline, thus facing elevated preference-leakage risk. For example, in our experiments, the Vicuna series—trained via GPT-4-distilled data—is later evaluated by GPT-4-turbo, naturally forming a within-family relationship between the generator and judge [1]. We agree that the case where architectures diverge but pre-training corpora overlap (e.g., Mistral vs. Gemma) represents an interesting intermediate scenario. We plan to include an additional discussion of this mixed-relatedness case in the camera-ready version, clarifying where architectural versus corpus overlap may contribute to leakage strength.
>
>
> **On Significance Testing for PLS.** We thank the reviewer for this suggestion. Following your comment, we highlight that our appendix already provides variance and bootstrap significance tests. The results are summarized below:
>
> | Judge Pair | Student | PLS (%) | Significance |
> |-------------|----------|---------|---------------|
> | GPT-4o vs Gemini-1.5 | Mistral-7B | **28.5** | *** (p < 0.001)** |
> | GPT-4o vs LLaMA-3.3 | Mistral-7B | −1.1 | n.s. |
> | LLaMA-3.3 vs Gemini-1.5 | Mistral-7B | 7.4 | ** (p < 0.01)** |
> | GPT-4o vs Gemini-1.5 | Qwen-2.5-14B | **37.9** | *** (p < 0.001)** |
> | GPT-4o vs LLaMA-3.3 | Qwen-2.5-14B | 1.2 | n.s. |
> | LLaMA-3.3 vs Gemini-1.5 | Qwen-2.5-14B | **26.3** | *** (p < 0.001)** |
>
>
> | Model Pair | Mean (ArenaHard) | Var (ArenaHard) | Mean (AlpacaEval 2.0) | Var (AlpacaEval 2.0) |
> |-------------|------------------|----------------|-------------------------|----------------|
> | Mistral-GPT-4o vs Mistral-Gemini-3.3 | **28.67** | 0.063 | **19.20** | 0.490 |
> | Mistral-GPT-4o vs Mistral-LLaMA-3.3 | 0.50 | 0.910 | 0.20 | 1.240 |
> | Mistral-LLaMA vs Mistral-Gemini | 12.93 | 0.583 | 19.87 | 0.013 |
>
> These results show that the main positive PLS findings (e.g., GPT-4o & Gemini) are statistically significant (p < 0.001), while low or negative cases (e.g., *GPT-4o & LLaMA-3.3*) are not. In the revised manuscript, we will add confidence intervals and explicitly reference these tables in the main text to reinforce the statistical robustness of our findings.

---

> > ### Author Response · Authors · 2025-11-22
> > **Response (part-2)**
> >
> > **DPO’s Leakage-reduction Effect.** We agree that distinguishing these effects is important. In our experiments, DPO and SFT were trained on identical datasets and with the same number of optimization steps. We also monitored training-loss convergence to ensure comparable learning stability. Therefore, the observed reduction in PLS for DPO cannot be attributed to under-training or reduced capacity. Prior work [2] also demonstrates that DPO reduces memorization relative to SFT, thereby mitigating contamination-like effects. Combined with our empirical results (average PLS ≈ 5.2 % for DPO vs 23.6 % for SFT), we believe the conclusion that DPO intrinsically suppresses preference leakage by discouraging over-imitation of generator style is well-supported and reliable.
> >
> > **On Combining Contextual Calibration with Adversarial or Paraphrased Prompting.** Indeed, these approaches are complementary. As described in Section 5.7 and Appendix C.4, our mitigation methods can be grouped into two categories:
> >
> > | Category | Methods | Core Function |
> > |-----------|----------|----------------|
> > | **Input- or reasoning-level debiasing** | Prompting, Chain-of-Thought (CoT), Paraphrasing | Reduces bias during input and inference by altering prompts or reasoning chains |
> > | **Output-level calibration** | Auto-Calibration, Contextual Calibration | Adjusts evaluation scores post-hoc to offset systematic bias |
> >
> > Combining these two layers is both feasible and promising: for instance, adversarial or paraphrased prompting could first reduce lexical overlap, while contextual calibration could then dynamically correct residual scoring skew.  We will extend this discussion in the revised version, outlining joint-application strategies as an important future direction for mitigating preference leakage.
> >
> >
> > **For Other Suggestions.** We sincerely appreciate these constructive ideas. Expanding our study to include additional commercial LLM families such as DeepSeek and Grok, evaluating leakage on affective or sentiment-analysis benchmarks, and exploring cross-family transfer (e.g., DeepSeek & Grok & Gemini) would indeed deepen our understanding of preference leakage’s generalization patterns.  We view these as valuable next steps and plan to investigate them in future work to enrich the conceptual connection between preference as a behavioral bias and its affective correlates in evaluation settings.
> >
> > We hope our response addresses the reviewer’s concerns and will be open to any further discussion!
> >
> > [1] Zheng, Lianmin, et al. "Judging llm-as-a-judge with mt-bench and chatbot arena." Advances in neural information processing systems 36 (2023): 46595-46623.
> >
> > [2] Hayes, Jamie, et al. "Measuring memorization in RLHF for code completion." The Thirteenth International Conference on Learning Representations. 2024.

---

### Official Review · Reviewer_nucm · 2025-10-31

**Soundness:** 3
**Presentation:** 3
**Contribution:** 3
**Rating:** 6
**Confidence:** 3

**Summary:**

The paper focuses on "preference leakage," a contamination issue in the LLM-as-a-Judge paradigm. It occurs when related data-generator LLMs (M_G) and evaluator LLMs (M_J) (same model, inheritance, same family) lead M_J to favor student models (M_S) trained on M_G’s synthetic data, inflating scores.
Using 3 generation/evaluation LLMs (GPT-4o, Gemini-1.5, LLaMA-3.3), 2 student models (Mistral-7B, Qwen-2.5-14B), and 2 benchmarks (Arena-Hard, AlpacaEval 2.0), the authors propose Preference Leakage Score (PLS) to quantify bias. Experiments confirm widespread leakage,worse in smaller M_S, subjective tasks, and SFT. They also analyze factors like data mixing and test mitigation.

**Strengths:**

* The paper fills a gap in prior research on LLM-as-a-Judge biases (e.g., egocentric bias) by focusing on the subtle, synthetic data-mediated bias between related models, rather than simpler self-favoritism.

* The paper maintains good clarity in structure and expression. It logically organizes content from problem definition to experimental design, results, and mechanism analysis.

**Weaknesses:**

* Mechanistic Analysis Insufficiency: While the paper attributes leakage to "spurious features (style/format)" inherited by student models, it does not empirically validate these features.  It could use feature attribution methods (e.g., SHAP, LIME) to identify which specific stylistic/formatting elements (e.g., sentence structure, terminology) drive M_J’s bias, or conduct ablation studies (e.g., paraphrasing synthetic data to remove style) to test if leakage diminishes.

* Real-World Impact Evidence Weakness: The claim that preference leakage impacts leaderboards relies on a single case study (AlpacaEval 2.0) with limited model coverage (only Vicuna/GPT-4-related models). It lacks: 1) analysis of other major leaderboards (Chatbot Arena, MT-BENCH); 2) longitudinal data (e.g., whether model rankings shift when using unrelated M_J); 3) human validation (e.g., comparing biased LLM rankings to human rankings for leaked vs. non-leaked model pairs).

**Questions:**

Regarding quantitative indicators：
1. The formula defaults to a 50% fair win-rate for evaluators between two student models without leakage. However, inherent quality differences (architecture, training data) may cause deviations (a superior model naturally has a 60% win-rate). Lack of baseline calibration (e.g., human evaluations as the true benchmark) leads to misjudgment, attributing inherent quality gaps to preference leakage.
2. The formula assumes unrelated evaluators (J_unrelated) are unbiased. In reality, all LLMs may have common biases (length bias, style preference) that skew judgments. Failure to exclude these biases undermines the accuracy of the "normal preference" baseline in PLS calculation.

---

> ### Author Response · Authors · 2025-11-22
> **Response (part-1)**
>
> We sincerely thank the reviewer for the acknowledgement of our work. We would like to provide further clarification and explanation for the following points.
>
> **Effect of Spurious Feature on Preference Leakage** We thank the reviewer for this insightful comment. To further validate that spurious features (style, format, wording) contribute to preference leakage, we conducted an additional ablation experiment. Using Qwen-3-8B as the rewriting model, we applied a paraphrasing pipeline to selectively remove these features from model responses before evaluation. Because Gemini-1.5 is no longer available, we used Gemini-2.0 as the judge model. The table below shows the new PLS scores after removing each feature type. We observe significant PLS reductions in the two pairs that initially showed strong leakage (GPT & Gemini, LLaMA & Gemini), confirming that spurious features are key contributors. Among the three features, style and format removal produce the largest decrease, whereas wording has a smaller effect. Differences across model pairs suggest that distinct judges are sensitive to different spurious traits—e.g., GPT-based judges may favor stylistic similarity more than LLaMA-based ones. These findings empirically substantiate our mechanistic explanation of preference leakage. We appreciate the reviewer’s encouragement to deepen the analysis. The above ablation directly supports our earlier hypothesis that spurious stylistic and formatting patterns embedded in student models drive biased judgments. We will include this experiment and its detailed implementation (prompting design, paraphrasing strategy, and model setup) in the revision.
>
> | Setting | GPT & Gemini | GPT & LLaMA | LLaMA & Gemini |
> |----------|---------------|--------------|----------------|
> | Baseline | 17.5% | 2.3% | 18.8% |
> | – w/o style | 9.0% | 3.3% | 14.6% |
> | – w/o format | 9.8% | 1.9% | 14.5% |
> | – w/o wording | 11.2% | 2.4% | 18.2% |
>
>
> **Real-world Impact Analysis.** We appreciate the reviewer’s suggestion to broaden leaderboard analysis. Practically, however, there are very few publicly documented student–teacher (distillation) pairs; most LLMs do not disclose which generator they were distilled from, which makes systematic expansion across many model pairs very difficult. Additionally, many community leaderboards are hard to align with our LLM-as-a-judge setup: they either do not cover the same set of models, use heterogeneous scoring protocols, or lack the metadata needed for controlled comparisons. For these reasons we focused on AlpacaEval and LMArena (Chatbot Arena) as meaningful, interpretable case studies. Furthermore, re-evaluating every leaderboard entry with alternate judges or collecting longitudinal rank shifts is computationally and logistically expensive at the scale of major leaderboards. Running such large-scale multi-judge re-evaluations for all models would substantially increase cost and take time beyond this submission. In all, we agree these analyses would strengthen external validity. We will explicitly state these practical constraints in the revision, and pursue lighter-weight follow-ups in future work.
>
>
> **Assumption on 50-50 Win-rate.** We appreciate the reviewer calling attention to the implicit interpretation of a “fair” 50% win-rate. To be explicit: PLS does not assume that the true, unbiased win-rate between two student models must be 50/50. Instead, PLS measures relative distortion introduced by judge-relatedness: it quantifies how much each judge’s reported win-rate for their related student deviates from that judge’s local average across the paired comparisons (see Section 4 and PLS definition). Concretely, suppose the true (human) win-rate between Student A and Student B is 40/60, and Judge A reports 50/50 while Judge B reports 30/70; the gap between each judge’s reported rate and the corresponding local AVG still produces a positive PLS reflecting leakage toward the related student. Put differently, PLS captures asymmetric shifts in judges’ reported preferences caused by relatedness, not absolute departures from an assumed 50/50 ideal.

---

> > ### Author Response · Authors · 2025-11-22
> > **Response (part-2)**
> >
> > **Assumption on Unbiased Evaluator.** We thank the reviewer for pointing out this potential misunderstanding. Our formulation in Equation (1) requires that the alternative judge \( M_{J'} \) be *not related* to the data generator \( M_G \) and possess *comparable intrinsic assessment capability* to \( M_J \), but it does not assume that \( M_{J'} \) is perfectly unbiased.
> >
> > Formally, in our definition,
> >
> > $$
> > E_{x,y_S \sim P_{M_S}} [ S_{M_J}(y_S|x) \mid M_G \sim_{rel} M_J ]  > E_{x,y_S \sim P_{M_S}} [ S_{M_{J'}}(y_S|x) \mid M_G \not\sim_{rel} M_{J'} ]
> > $$
> >
> > we only require that \( M_{J'} \) has similar evaluation capacity, not that it provides an absolutely unbiased baseline.
> >
> > In practice, LLM judges inevitably share common biases—such as length preference or stylistic tendencies—which we consider as general evaluator noise. Crucially, *preference leakage* refers to a **systematic bias correlated with model relatedness**, while unrelated biases that do not depend on \( M_G \) act as random noise and therefore do not affect the relative deviation captured by the Preference Leakage Score (PLS).
> >
> > In other words, if a judge \( M_J \) exhibits a general bias independent of \( M_G \), this bias remains approximately constant across students and will cancel out when computing PLS.
> >
> > We hope our response addresses the reviewer’s concerns and will be open to any further discussion!

---

> > > ### Comment · Reviewer_nucm · 2025-11-26
> > > **comment**
> > >
> > > Thanks for your response. Most concerns are addressed, and i think 6 is a reasonable score. Good luck.

---

> > > > ### Author Response · Authors · 2025-11-26
> > > > **Thanks for your time and effort**
> > > >
> > > > We really appreciate the reviewer's valuable feedback and are glad to see our further response and explanation address most concerns raised initially.

---

### Official Review · Reviewer_F5m9 · 2025-10-31

**Soundness:** 3
**Presentation:** 3
**Contribution:** 2
**Rating:** 6
**Confidence:** 4

**Summary:**

This paper introduces and defines "preference leakage", a novel and subtle contamination problem in the LLM-as-a-judge paradigm. Preference leakage occurs when the data-generating LLM ($M_G$) and the evaluating LLM ($M_J$) are "related", for example, being the same model, having an inheritance relationship (e.g., fine-tuning), or belonging to the same model family (e.g., GPT-3.5 and GPT-4).

**Strengths:**

The primary strength is the paper's originality. The formalization of "preference leakage" as a distinct contamination vector, separate from both traditional data leakage and simple egocentric bias, is a novel and important conceptual contribution. Also, the empirical work is high-quality. The authors conduct a thorough and "full-factorial" investigation across multiple generators, students, and benchmarks . The further analyses (Section 5) are comprehensive and anticipate many of the reader's questions, such as the impact of data mixing ratios , relatedness types , and learning methods (SFT vs. DPO vs. ICL) .

**Weaknesses:**

1. The paper's main weakness is the preliminary and somewhat disconnected nature of the mitigation analysis (Section 5.7). While the paper excels at diagnosing the problem, the treatment section feels like an add-on. The mitigation experiments use a different setup (new datasets like PPE/MTBench, new "Error Bias" metric) than the main experiments (Arena-Hard/AlpacaEval 42, $PLS$ metric). This makes it hard to connect the findings. For example, how does the best mitigation (Contextual Calibration) affect the $PLS$ score on the main paper's task? This is not answered.

2. The paper explicitly defers a full solution: "We leave the exploration of detection, mitigation and calibration methods of preference leakage for future works". While this is fair, it does make the paper feel somewhat incomplete. The community is now aware of a major problem but is not given a validated, well-integrated solution within this same work.

**Questions:**

1. Could you apply your most promising mitigation method, Contextual Calibration, to your main experimental setup from Section 4? Specifically, how much does this method reduce the $PLS$ score (e.g., in the "Mistral-7B, GPT-4o & Gemini-1.5" case from Table 1, which was 28.7%)? Showing a direct reduction in your primary metric would make the contribution feel much more complete.

2. You cleverly show that while LLM judges cannot recognize their students, a fine-tuned BERT classifier can. This is a great finding. Have you considered using this classifier as a detection or mitigation tool itself? For example, could the classifier's confidence score for "relatedness" be used as a penalty term in the evaluation, or to flag high-risk comparisons for human review?

---

> ### Author Response · Authors · 2025-11-22
>
> We sincerely thank the reviewer for the acknowledgement of our work. We would like to provide further clarification and explanation for the following points.
>
> **Experiment Design Principles.** We appreciate the reviewer’s insightful observation regarding the difference between the main and mitigation experiment setups. This choice was deliberate and stems from the distinct purposes of these experiments. In our main experiments (Section 4), we aimed to obtain controlled and interpretable insights into preference leakage itself: quantifying the phenomenon across models. Hence, we designed a controlled experimental setting in which model architectures, data size, and fine-tuning protocols were strictly aligned, and all bias measurements were computed automatically via our proposed Preference Leakage Score (PLS). In contrast, Sections 5.4 and 5.7 focus on impact and mitigation analysis, where realism and external validity are prioritized. For this purpose, we adopted human-labeled benchmarks such as PPE and MT-Bench and introduced new evaluation metrics (Ranking Difference and Error Bias) suited to labeled data. These settings more closely resemble practical “LLM-as-a-judge” scenarios—e.g., Vicuna-style distillation from GPT-4—and thus enable an initial exploration of mitigation feasibility under real-world conditions. While the availability of labeled data prevents directly computing PLS in this context, our design maximizes complementarity between controlled and realistic analyses. We will further clarify this motivation and connection between sections in the revised version.
>
> **Mitigation Methods with External Detector.** We thank the reviewer for this excellent suggestion. We agree that leveraging an external detector to identify or calibrate preference leakage is a highly promising research direction. As the reviewer notes, one straightforward approach would be to incorporate the detector’s confidence (or “relatedness” score) as a penalty or calibration term during evaluation to attenuate biased judgments. Beyond that, we also envision a multi-agent evaluation setting where detector-provided signals are integrated with other indicators—such as each judge-agent’s reliability or performance score—to determine aggregation weights for individual judgments on a per-sample basis. This hybrid approach could enhance robustness in ensemble or committee-based evaluation frameworks. We have added a brief discussion of this idea in the revised rebuttal version and plan to expand on it more comprehensively in the camera-ready version.
>
> We hope our response addresses the reviewer’s concerns and will be open to any further discussion!

---

> > ### Comment · Reviewer_F5m9 · 2025-11-26
> >
> > Thank you for your response. I believe the initial scores are appropriate.

---

> > > ### Author Response · Authors · 2025-11-26
> > > **Thanks for your time and effort**
> > >
> > > We really appreciate the reviewer's valuable feedback and are open to further discussion on any points the reviewer may have concerns about during the response period.

---

### Comment · Area_Chair_xyPA · 2025-11-25

Dear Authors and Reviewers

i would like to thank the authors for their replies and i would like to encourage the reviewers to read the authors response and the other reviews. Please reply early to the authors response to leave enough time for the discussion.

Best regards
AC

---

### Meta-Review · Area_Chair_exgj · 2026-01-03

**Summary:**

All reviewers originally gave a positive score: 6668. This is clearly above the acceptance threshold.

**Reviewer Concerns:**

The reviewers were concerned about the disconnected nature of the mitigation analysis, the limited depth of mechanistic evidence identifying which features caused leakage, and the weak evaluation (limited leaderboards and models). They also raised concerns about the assumptions and confidence intervals. The authors responded in their rebuttal with additional results and clarifications, noting, for example, that the appendix already included some of the requested results.

**Reviewer Scores:**

I think all reviewers would have maintained their scores if the incident hadn't happened. Two reviewers (score: 6) explicitly said they would maintain their scores. Reviewer fJqb (score: 6) noted that the rebuttal does not fully resolve their concerns. Reviewer HBz8 already gave a high score of 8, and the authors did not add the requested model experiments in the rebuttal.

---

### Decision · Program_Chairs · 2026-01-26

Accept (Poster)